# Persistence of Mating Suppression of the Indian Meal Moth *Plodia Interpunctella* in the Presence and Absence of Commercial Mating Disruption Dispensers

**DOI:** 10.3390/insects11100701

**Published:** 2020-10-14

**Authors:** Leanage K. W. Wijayaratne, Charles S. Burks

**Affiliations:** 1Department of Plant Sciences, Faculty of Agriculture, Rajarata University of Sri Lanka, Puliyankulama, Anuradhapura 50000, Sri Lanka; kanaka.wijayaratne@gmail.com; 2USDA, Agricultural Research Service, San Joaquin Valley Agricultural Sciences Center, 9611 South Riverbend Avenue, Parlier, CA 93648, USA

**Keywords:** stored products pest, Indian meal moth, *Plodia interpunctella*, mating disruption

## Abstract

**Simple Summary:**

A novel assay system was used to facilitate replicated studies examining the impact of commercial mating disruption dispensers on *Plodia interpunctella*. Both direct and indirect exposure to passive mating disruption dispensers for as little as 2 h suppressed mating throughout the rest of a 10 h scotophase. This is the first direct evidence that for *P. interpunctella*, transient exposure to commercial mating disruption dispensers is sufficient to suppress male orientation to females without re-exposure to the mating disruption dispensers. An improved understanding of mechanisms for mating disruption can improve both development of future products and how current products are used.

**Abstract:**

The Indian meal moth *Plodia interpunctella* (Hübner) (Lepidoptera: Pyralidae), is controlled by commercial mating disruption dispensers using passive release to emit high concentrations (relative to females or monitoring lures) of their principal sex pheromone component, (9Z,12*E*)-tetradecadienyl acetate. Since *P. interpunctella* is sexually active throughout the scotophase, an assay system was developed to determine the importance of direct interaction of the male with the dispenser, and whether exposure to mating disruption early in the night is sufficient to suppress mating throughout the night. Exposure to mating disruption dispensers in the mating assay chamber for the first two hours of a 10-h scotophase significantly reduced mating when females were introduced four hours later. Mating was also reduced to a lesser degree in a concentration-dependent manner based solely on re-emission of pheromone, and when males were exposed outside the mating assay chamber. These results indicate that the commercial mating disruption dispensers can suppress mating throughout the night based on interaction with the dispenser early in the night. Desensitization resulting from attraction to a high-concentration pheromone source is important to this suppression, but other factors such as re-emission from the environment may also have a role. These observations imply a non-competitive mechanism for *P. interpunctella* with the product studied, and suggest that effectiveness of the mating disruption dispenser might be augmented by using them in conjunction with another formulation such as an aerosol or micro-encapsulated product.

## 1. Introduction

Semiochemical control methods have become an increasingly important part of insect pest management over recent decades [1,2]. Mating disruption is one of several pest management strategies that use pheromones or other semiochemicals for control of insect pests [3]. Other strategies include mass trapping [4], attract-and-kill [5], and push-pull strategies [6]. Mating disruption mechanisms are categorized most broadly as either competitive or non-competitive [2,7]. Both competitive and non-competitive mating disruption may be accomplished by one or a combination of mechanisms; e.g., competitive attraction or induced arrestment for competitive disruption; or desensitization, camouflage, or sensory imbalance for non-competitive disruption [7]. Different formulations are used, such as aerosol, hand-applied dispensers, or micro-encapsulated flowable products [2,7]. The concentration and presentation of pheromone by mating disruption dispensers can affect the mechanism by which mating disruption works, as demonstrated with the oriental fruit moth *Grapholita molesta* (Busck) [8]. For orchard and horticultural pests, absorption and re-emission of pheromone from mating disruption dispensers are apparently important factors for some species [9,10], but not for others [11]. Some reviews of mating disruption have called attention to a trend of increased area of cultivated or forest land treated using this technique [1,7]. Most well-established mating disruption products target lepidopteran pests [12].

Stored product pest management is difficult to compare with use in agronomic, horticultural, or forestry applications because treatment is generally to structures rather than land. Mating disruption and other semiochemical control methods have, nonetheless, also increased in importance in this area of pest management [13,14,15]. As in other applications (horticulture, agronomy, forestry, etc.), mating disruption for stored products pest is most developed for lepidopteran pests; in particular, for a group of stored products pests of the subfamily Phycitinae. The Indian meal moth *Plodia interpunctella* (Hübner), the almond moth *Cadra cautella* (Walker), and the Mediterranean flour moth *Ephestia kuehniella* (Zeller) are three widely distributed pests of preprocessed and finished durable commodities that share the same primary sex pheromone component, and among which cross-attraction occurs [16].

Research has examined a variety of semiochemical control techniques for these species, including mass trapping [17,18,19], attract-and-kill [20,21], and mating disruption [14,15]. Research on mating disruption for stored product Phycitinae has included use of hand-applied (reservoir) dispensers [22,23,24,25], aerosol dispensers [21,26,27], and micro-encapsulated formulations [28]. A laboratory wind tunnel study found that exposure to increasing concentrations of a sex pheromone blend raised the lower as well as upper threshold of the response of the almond moth [29]. Response of Indian meal moth males to unmated females and monitoring lures of varying strength in the presence or absence of commercial mating disruption [26] suggest that a similar phenomenon occurs in that species. The calling period is longer for the Indian meal moth than for the almond moth or for the Mediterranean flour moth [30], so a longer desensitization period for this species will be important for successful mating disruption. A formulation containing only the principal component, (9*Z*,12*E*)-tetradecadienyl acetae (Z9,E12-14:OAc) suppresses mating by *P. interpunctella* as effectively as more complete blends [31], so commercial products use single-compound formulations from hand-applied dispensers. Several studies have demonstrated efficacy for such formulations for control of *P. interpunctella*, *C. cautella*, and *E. kuehniella* in commercial facilities [32,33,34,35,36].

In the present study we test whether prior exposure of *P. interpunctella* males to a commercial hand-applied mating disruption dispenser causes subsequent suppression of mating. Replication is obtained inside 208 L experimental chambers. Preliminary experiments examine whether the experimental arena is sufficiently large that mating occurs by response to the pheromone as opposed to random contact. An initial mating disruption experiment compared the impact of exposure to mating disruption dispensers during only initial parts of the night with males exposed all night or not exposed. A subsequent mating disruption compared mating in chambers used in the first experiment, to determine if there were residual effects from the hand-applied dispenser introduced into and then removed from the chambers. A final experiment examined the impact of exposure outside the experimental chamber on subsequent mating inside the chamber. These provide information on the extent to which close contact to dispensers is important to their effectiveness, and whether continued interaction with these dispensers during the night is necessary for their effectiveness.

## 2. Materials and Methods

### 2.1. Insects and Mating Disruption Dispensers

Adults were from a laboratory colony of *P. interpunctella* established from moths emerged from a load of infested pistachios received from Kern County in 2015, and maintained in the laboratory for 50 to 60 generations at the time of the experiments. The colony was maintained on wheat bran diet (wheat brain, brewer’s yeast, honey, and glycerin) [37] at 26 °C, 60% r.h. (relative humidity), under a photoperiod of 14:10 (L:D) with photobase beginning 06:00 local time (UCT-7). Rolled bands of corrugated cardboard (1 × 5 cm) were placed in the 4 L glass culture jars to provide a pupation site to wandering larvae. Males and females were separated from the cardboard rolls in the pupal stage based on the morphology of the ventral terminal segments [38,39]. The mating disruption dispenser used was Cidetrak IMM-36 (Trece Inc., Adair, OK, USA). These polyvinyl carbonate dispensers are approximately 10 × 2.5 cm and (per the label) contain 160 mg Z9,E12-14:OAc per dispenser.

The release rate of these devices was estimated by quantification of volatile collection using gas chromatography. Volatiles from mating disruption dispensers were collected using a borosilicate glass tube (4 mm I.D. (inner diameter) × 8 cm length) packed with 60 mg of SuperQ^®^ (Alltech Associates, Deerfield, IL, USA) adsorbent which was fitted to a volatile collection chamber (4 cm I.D. × 30 cm length) purchased from Analytical Research Systems Inc. (Gainesville, FL, USA). Laboratory house air was controlled by an upstream needle-valve flow meter that delivered air to the head of the volatile collection chamber packed with Supelpak™-2 adsorbent for air purification (Supelco, Bellefonte, PA, USA). The air flow was confirmed using an Agilent flow through meter (Agilent Technologies) (Santa Clara, CA, USA). Volatile collections were made at an airflow of 500 mL/min at room temperature (24–25 °C) for 1 h. Z9,E12-14:OAc was subsequently eluted from the SuperQ trap using gravity and 3 mL of hexane into glass vials to which 3 µg of the internal standard were added, and 1 μL was analyzed on GC-MS (gas chromatography-mass spectroscopy). The internal standard nonadecane (C19) was purchased from Tokyo Chemical Industry Co. (TCI America, Portland, OR, USA). A stock solution of the internal standard was prepared gravimetrically in hexane at a concentration of 1 µg/µL. Analyses were performed on a Hewlett Packard model 6890 gas chromatograph equipped with a cool on-column injector and a 5975C MSD detector in EI (electron ionization) mode (Agilent Technologies, Santa Clara, CA, USA) and a 30 M, 0.32 mm I.D., 0.25 µm film thickness Stabilwax^®^-DA analytical column (Restek Corp. Bellefonte, PA, USA). Carrier gas (ultra high purity He) was held at a constant flow (1.5 mL/min). Analyses were conducted in EI mode using selected ion monitoring scanning the five most abundant ions for Z9,E12-14:OAc and C19 respectively listed in order of most abundance: ZETA (m/z 67.1, 79.1, 43, 55.1, 95.1); C19 (m/z 57.1, 71.1, 85.1, 43.1, 99.1). The following GC temperature program was employed: injection temperature of 50 °C with a one-min hold, a temperature increase of 10 °C per min to 200 °C with a 2-min hold, then a temperature increase of 50 °C per min to 240 °C with a 8-min hold. Volatile collections were made from 5 mating disruption dispensers.

### 2.2. Assay Procedures

Experimental mating chambers were formed from high density polyethylene barrels (S-9945NAT, Uline, Pleasant Prairie, WI, USA). These chambers were white cylinders (partially translucent) and had inside dimensions of 50 cm diameter × 90 cm length and an internal volume of 208 L. A lid for the chamber was held in place with a metal band that could be sealed or release with a lever. This lid had a 5 cm circular port 5 cm from one edge. This part was closed with an appropriately sized black rubber stopper. When in use, the chamber was placed on its long side with the port in the lid positioned vertically opposite of the bottom along which the container rested. The lid was carefully removed to allow moths to be introduced, and then replaced. The mating disruption dispensers were suspended by hanging them on the hook on the end of a string approximately 60 cm long with a hook on each end. The dispenser was suspended on one hook approximately 20 cm below the port, and this was held in place by closing the stopper over it. The other end of the string was left hanging outside the chamber to facilitate retrieval of the mating disruption dispenser from the chamber.

Experiments were conducted inside a manufactured structure (Northwest Building Systems, Boise ID USA) with floor dimensions of 2.79 × 12.2 m, and a ceiling height of 2.56 m (area 34 m^2^; volume 87 m^3^). Coverings over the windows and a partition by one of the doors was used to prevent the entry of external light, and a programmable light timer (ST01, intermatic.com) was used to select the photoperiod and scotoperiod. An external programmable thermostat (T1100RF, Venstar, https://venstar.com/thermostats/wireless/) was used to maintain the temperature between 22 and 24 °C. Relative humidity was maintained at 60% with a portable humidifier with an internal water volume of 23 L (AirCare 831000, Essick, Little Rock, AR, USA). The chambers were held in racks in groups of four, with two at floor level and two stacked above this bottom level. One rack of four chambers was along the short wall, and the other 16 along one long wall. Assays were conducted under a reversed 14:10 L:D with scotophase beginning at 17:30 local time.

Assays were performed using adults emerged from pupae reared and sexed as described in the previous section. Sexed pupae were then placed inside 946 mL jars in a reach-in incubator under the experimental reversed photoperiod and maintained at 60% r.h. Adults were used in experiments 2–4 days after eclosion. At the appropriate age adults were gently aspirated, release into a cube-shaped wire mesh sleeve cage (30 × 30 × 30 cm), and collected in groups of four males or four females into clear 28 mL polycarbonate vials which were used to transfer moths to the structure containing the experimental chambers. Males were transferred into the chambers through the ports by shaking them out of the 28 mL polylcarbonate vials. Males were introduced to the chambers shortly before the beginning of scotophase, and females were introduced six hours later with the aid of head lamps with red LED lights. At the end of scotophase the barrels were turned upright and opened. The adults were removed using a hand-held vacuum aspirator (InsectaVac 2809B, BioQuip, Rancho Dominguez, CA, USA). The moths were quickly anesthetized by placing vials from the aspirator in a 473 mL glass jar with two ports in the lid and flushing this jar with carbon dioxide. Quick inactivation avoided additional mating after removal from the chambers due to the moths being crowded closely together. After they became inactive the moths were transferred to 16 × 100 glass tubes with lids. The tubes were further flushed with carbon dioxide before transferring them to a freezer where the tubes were held at −20 °C until females could be dissected. Females were scored as mated or unmated based on the presence or absence of a spermatophore in the bursa copulatrix at the time of dissection.

### 2.3. Characterization of the Assay

Assays in light or dark conditions, and in the moths’ entrained photoperiod or scotoperiod were used to compare diel patterns and photoperiodic influences of mating for three different arena sizes: the 28 mL polycarbonate vials used to isolate moths, 946 mL jars with wire mesh lids, and the 208 L assay chambers described above. Assays in the 28 mL vials were conducted by pairing moths 2 h after the start of their entrained scotoperiod or photoperiod, and in light or dark conditions. For these trial individuals, rather than groups of four adults, were gathered in the vials and paired by placing a male and a female in the same container. Mating for the 28 mL arena was scored based on observation of courtship and copulation between the paired moths. Moths for the 946 mL assays were paired in light conditions shortly after the beginning of their entrained photophase, or in dark conditions shortly after the beginning of scotophase. The moths in these 946 mL arenas were observed initially, then left until the end of the photophase or scotophase and then dissected to determine the presence or absence of a spermatophore. To compare the 208 L assay with the two smaller arenas, groups of 4 males were introduced at the beginning of the photophase (6 barrels) or scotophase (6 barrels), and groups of 4 females were added 6 h later. Subsequent steps in these barrel assays were as described in the previous section.

### 2.4. Experiment 1: Effect of Time and Duration of Exposure on Mating

An initial experiment examined the impact of the time males were exposed to mating disruption dispensers and the time between when dispensers were removed, and females were present on mating (Figure 1). There was a total of five treatments:

(1) Males were exposed to mating disruption dispensers for two hours, and were provided four hours after removal of the dispensers for recovery prior to the introduction of females into the barrels.

(2) Males were exposed to mating disruption dispensers for four hours, and were provided two hours after removal of the dispensers for recovery prior to the introduction of females into the barrels.

(3) Males were exposed to mating disruption dispensers for six hours, and the dispensers were removed immediately prior to the introduction of females into the barrels.

(4) Mating disruption dispensers were present in the barrel throughout the 10 h scotophase in which the experiment was conducted.

(5) No mating disruption dispensers were present in the barrel at any time during the experiment.

This experiment was conducted with four replicate blocks, and three separate dates for a total of 12 replicates (with the treatment chamber as the experimental unit). Observations in which four females were not recovered were discarded and, due to escaped moths, the actual number of replicates for the 0, 4, and 10 h treatments were 11.

### 2.5. Experiment 2: Effect of Prior Pheromone Treatments in Chambers on Mating in the Absence of Dispensers

A subsequent experiment examined whether there was an effect of mating solely from any pheromone that may have been absorbed and re-emitted by the chambers themselves. The treatments were as in the previous experiments, except that no mating disruption dispensers were placed in any barrel and the explanatory variable was the treatment that the treatment chamber received previously. This experiment was conducted one to two days after the previous experiment. The experiment was conducted two times with the four replicate blocks for eight replicates per treatment.

### 2.6. Experiment 3: Effect of Exposure of Males to Pheromone Dispensers Outside the Assay Chamber on Mating

Following the previous experiment, the chambers were allowed to air out for two weeks, and then the mating assay involving introduction of females four hours before the end of the scotophase was repeated with males treated by exposure to mating disruption outside the chambers for four hours, or sham-treated with no exposure to mating disruption. These were the equivalent of treatments 2 and 5 (Figure 1), but with exposure to mating disruption outside the 208 L assay chamber. These treatments were randomized within the previous exposure of the mating chamber; i.e., males exposed to mating disruption were randomly assigned to two of the chambers previously used for untreated controls, and the other two had sham-treated males. The same procedures were used for the chambers as used for two-, four-, and six-hour treatments in previous experiments. Mating disruption exposure was re-randomized between the previous exposure levels between the three replicates in time. The 10 h treatment chambers were used only on the third iteration, and there was an escape from one of these. There were thus a total of 26 replicates for the sham-treated males, and 25 replicates for the males treated with mating disruption.

To allow dissipation of absorbed pheromone, chambers were removed from the structure where they had been housed, placed outside in the open. They were turned so that the open side was on the top rather than the side, and left that way for two weeks. The lids were placed outside separately, with the inside turned up. The local weather at the location (Parlier, CA, USA) was seasonally normal; i.e., no clouds and daytime highs averaging 31 °C.

Exposure of the males and sham treatment were conducted in 0.61 × 0.61 × 0.61 m aluminum mesh sleeve cages (#1450D, Bioquip Products, Rancho Dominguez, CA, USA). These cages had access through a 0.3 × 0.3 m cloth mesh sleeve on one corner of the cage, and had a solid aluminum floor. The cage used for mating disruption treatment was in a manufactured structure identical and adjacent to the one containing the mating chambers, and the cage used for sham treatment was in the same structure as the barrels, on an opposite wall. Treatment was administered by placing a Cidetrak IMM-36 dispenser on a piece of aluminum foil in the center of the cage, and releasing males from 28 mL vials into the cage at the beginning of scotophase. After four hours of exposure, males were collected back into the 28 mL vials. The vials were collected into an open-topped box with was then shrouded with a thick black vinyl tarp as the males were moved from the dark structure where mating disruption treatment was administered to the dark structure containing the mating chambers. The same procedure was performed with the sham-treated males; i.e., after 4 h in the aluminum mesh cage, they were recollected, shrouded, taken to the door of the adjacent structure, and returned.

### 2.7. Data Analysis

Data were summarized and illustrated using R 4.0 [40] in RStudio 1.4.5 [41] with tidyverse 1.3.0 [42]. Fisher’s Exact test (used to examine impact of light and entrainment on mating in different arena sizes) and chi-square tests of contingency tables (illustration of impact of random effects in experiment 3, exposure outside chamber) were calculated using base R. Analysis using generalized linear models (GLM) or generalized linear mixed models (GLMM) was performed in the SAS (full name) System [43], respectively using the Proc GENOD (full name) and Proc GLIMMIX. For the first two experiments in the mating disruption chambers (time of exposure to the dispenser and effect of prior treatment), individual chambers were a repeated measure. Fisher’s exact test was also used to compare the proportion of mated females between the two days for experiment 2. For the third experiment in the mating chamber (exposure outside the chamber), the prior pheromone treatment of the chamber and the iteration (replicate in time) were random factors. In all cases a binomial error distribution was used. Differences between treatment means were examined using the least-squared means procedure (LSMEANS). In addition, percent mating suppression index was used to compare between experiments: (100[1 − treatment/control]) where treatment and control are respectively the proportion of moths mated in a treatment in the untreated control.

## 3. Results

### 3.1. Characterization of the Assay

When *P. interpunctella* was paired in 28 mL or 946 mL arenas, mating was almost immediate, 66–100% of the pairs mated (Appendix A), and there was no significant difference in the proportion of pairs mated between vials held under dark or light conditions (Appendix A). In contrast, significantly more females were mated in the 208 L mating chambers when the assay was conducted during scotophase compared to when it was conducted under photophase (Figure 2).

### 3.2. Experiment 1: Effect of Time and Duration of Exposure on Mating

The release rate of the mating disruption dispensers used in this study was estimated by as 1.25 ± 0.032 µg Z9,E12-14:OAc per hour over a 24-day period (*n* = 5). When males were exposed to a mating disruption dispenser inside the mating assay chamber, all mating disruption treatments resulted in significantly lower mating compared to the control (GEE, *Z* = 9.12, *p* < 0.0001) (Figure 3). The 6-h treatment in which males had no time away from the mating disruption dispenser resulted in significantly less mating than the 2 and 4 h treatments in which males had respectively 4 and 2 h without the presence of the mating disruption dispenser before the introduction the females. In contrast, suppression of mating was not significantly greater when the dispensers were left in place through the remainder of the scotophase after the addition of females. The suppression of mating in the chambers treated 2, 4, 6, and 10 h with mating disruption dispensers was respectively 50, 65, 76, and 81% with respect to the untreated control chambers.

### 3.3. Experiment 2: Effect of Prior Pheromone Treatments in Chambers on Mating in the Absence of Dispensers

Tests with the mating in the chambers shortly after the mating disruption treatments also found significant and dose-dependent reductions in mating based on the length of time that the barrels were exposed to mating disruption in the previous experiment (*Z* = 3.2, *p* = 0.0014) (Figure 4). The proportion of females mated in chambers exposed for two hours was not significantly less than that mating in the untreated controls, while the chambers exposed for 4 or more hours all had lower mating than the 0 or 2 h treatment (Figure 4). The suppression of mating in the chambers previously treated for 2, 4, 6, and 10 h treatments was respectively 28%, 44%, 67%, and 67% with respect to the chambers never treated with mating disruption. The number of females mated across all treatments was not significantly greater on the second day of experiment 2 than on the first day (Appendix A).

### 3.4. Experiment 3: Effect of Exposure of Males to Pheromone Dispensers Outside the Assay Chamber on Mating

Treatment of males in a cage outside the assay chamber resulted in a statistically significant 36% reduction of mating (Figure 5). There was no significant effect from the iteration (Figure 5a) or the previous pheromone treatment the chamber was exposed to in experiment 1 (Figure 5b). Within the conditions of the exposure, very few males made contact with the mating disruption dispenser. Most flew in an agitated manner in the center of the cage, hovering over the mating disruption dispenser. A few walked in an agitated zig-zag pattern on the floor of the cage, while wing-fanning, alternatively turning to and away from the pheromone dispenser.

## 4. Discussion

Initial experiments comparing the effects of circadian entrainment and light on mating in containers of different volumes demonstrated that the use of sex pheromone was important for location of mates in the 208 L mating chambers. Preliminary experiments comparing mating in the chambers and smaller arenas (28 mL vials or 946 mL jars) revealed that *P. interpunctella* mated when placed on the smaller arenas regardless of circadian entrainment or the presence or absence of light (Appendix A). This is consistent with previous observations that in stored product phycitine moths, the probability of mating is greatly increased by contact or near vicinity [22,44]. In contrast, the fact that males exposed in the assay chambers were significantly less likely to mate in light during their entrained photoperiod than in dark under their entrained scotoperiod suggests that this assay chamber was large enough that female-emitted sex pheromone was important for males to encounter females for mating. No multiple mating was found, presumably because the design of the assay involved exposure of equal number of virgin males and females for four hours.

The experiments in the mating chambers indicated that exposure to mating disruption dispensers was by itself sufficient to reduce mating when males were exposed to females hours later. However, both interaction of males with the dispenser and permeation of pheromone from the dispensers into the air and assay chamber surfaces were likely involved in suppression of mating in the first experiment, which was greater than the second experiment (suppression based on the length of exposure in the first experiment, presumably from re-emission) or the third experiment (exposure of males to mating disruption outside the mating assay chamber, therefore examining the effect on the male only). The fact that mating suppression was greater when mating disruption dispensers were presented in the assay chamber (65% suppression following a four-hour exposure) compared to only re-emission effects or only exposure outside the assay chamber (44% and 36%, respectively, for four-hour exposures) suggested that both direct exposure and re-emission contributed to the suppression seen in the first mating disruption experiment in the assay chamber.

The assay system in this experiment provides means of experimental replication, and is not representative of conditions conducive to control of stored product moths by mating disruption in the field. The total interior surface area of the assay chambers was 1.8 m^2^. Other studies have examined “wall and ceiling area” based on the observation that calling females are not observed on the floor [22,31]. If the lower quarter of the side of the cylinder is considered the bottom and subtracted, the interior area is 1.45 m^2^. The number of pairs per m^2^ was therefore 2.2–2.8, which is in the general range of density in other experimental studies of effects of pheromone release on mating suppression in stored product moths [21,24,27]. The presence of some mating in the barrels when moths were paired during the photoperiod suggests that a baseline level of mating in this assay comes from pairing following chance encounters (Figure 1). The difference in mating between treatments and the differences in responses between the three experiments using the assay system (Figure 2, Figure 3 and Figure 4) nonetheless provide cogent evidence that exposure to the dispenser early in the night is sufficient to provide night-long suppression, and the direct effects from interaction with the dispenser and indirect effects from re-emission are both factors in suppression of mating by these dispensers.

For some target species, the emission rate of pheromone from mating disruption dispensers can determine both the primary mechanism of mating disruption, and its effectiveness. For example, mating disruption in the *G. molesta* occurred by a competitive mechanism when performed in experimental enclosures using monitoring lures, but by a non-competitive mechanism when performed using commercial hand-applied (passive release) dispensers with a higher release rate [8]. A study of prior exposure of *C. cautella* to increasing concentrations of Z9,E12-14:OAc increased the optimum emission rate for reaching the emission source in wind tunnel tests [29]. Assays of *P. interpunctella* males with female-baited traps and with monitoring lures containing sequentially higher concentrations of Z9,E12-14:OAc suggested a similar effect in this species [45]. Experiments examining disruption of mating in *C. cautella* using an aerosol dispenser found that higher concentrations of Z9,E12-14:OAc were more effective [21]. A subsequent study of mating disruption for *P. interpunctella* found that an aerosol dispenser efficiently reduced mating [26]. For the former study the aerosol dispenser emitted pheromone onto a fabric pad to be re-emitted, whereas in the latter study the dispenser emitted directly into the atmosphere as is typical of aerosol dispensers used in orchards [46,47]. A study comparing disruption of mating in *C. cautella* using either micro-encapsulated sprayable products or a dense grid of monitoring-like dispensers formed from polyethylene microcentrifuge tubes found that both suppressed mating, but that polyethylene tube dispensers did so with a lower amount of pheromone [28]. Comparison of this previous study with the observations for *P. interpunctella* in the current study suggest that, as with *G. molesta*, the mechanism of mating disruption in stored product pyralid moths changes to non-competitive as the emission rate of mating disruption dispensers goes over a certain threshold. This would imply that mating disruption is less density-dependent under these circumstances.

The results of the current study suggest that for maximal effectiveness, *P. interpuctella* males must first be attracted to and approach these high-emission dispensers, after which they are unable to locate normal female-strength pheromone plumes for the rest of the night. The disadvantage of this approach is the possibility that males could encounter an authentic female plume before encountering a more distant but more concentrated plume from a mating disruption dispenser. This potential vulnerability is further increased by the nature of processing and storage environments which, unlike orchards, do not offer natural grids for regular spacing of dispenser. One plausible approach to address this vulnerability is to us relatively high-emission dispensers such as those used in the current study in conjunction with a lower-emission but more widely dispersed formulation. For example, a micro-encapsulated product [28] might be applied, possibly using aerosol insecticide application systems (aka. Foggers) [48]. A previous study indicated that mating disruption could save money by reducing the frequency of insecticide applications with such aerosol systems [26], but assessing the efficacy and economic viability of this suggested approach requires empirical studies.

## 5. Conclusions

Experiments in this study indicate that exposure of *P. interpunctella* males to commercial mating disruption dispensers outside of an assay chamber early in the night-long period of sexual activity suppressed ability of males to successfully mate with females for the rest of the night. Pre-treatment of assay chambers on prior nights was sufficient to suppress mating in males that never had direct access to dispensers, but greater suppression was observed when males did have direct access to mating disruption dispensers for part or all of the night. These findings provide evidence that male interaction with dispensers is important for effectiveness of these dispensers, and suggest that the primary mode of action is non-competitive via increased thresholds. This study thus provides greater insight into how existing treatments work, and suggests ways in which effectiveness might be further optimized.

## Figures and Tables

**Figure 1 insects-11-00701-f001:**
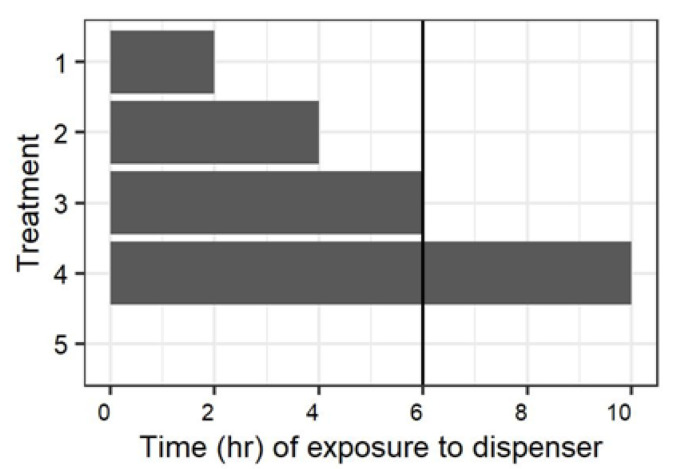
Mating disruption treatment regimens for experiments in 208 L chambers. Males were introduced into the chamber at the beginning of a 10 h scotophase, and females were introduced after six hours (vertical line). The first three treatments involved exposure to mating disruption dispensers inside the arena for the first two, four, or six hours before withdrawing the dispenser. The fourth and fifth treatment were respectively mating disruption dispensers left throughout the scotoperiod (including after the females were introduced), and no mating disruption dispenser in the arena at any time.

**Figure 2 insects-11-00701-f002:**
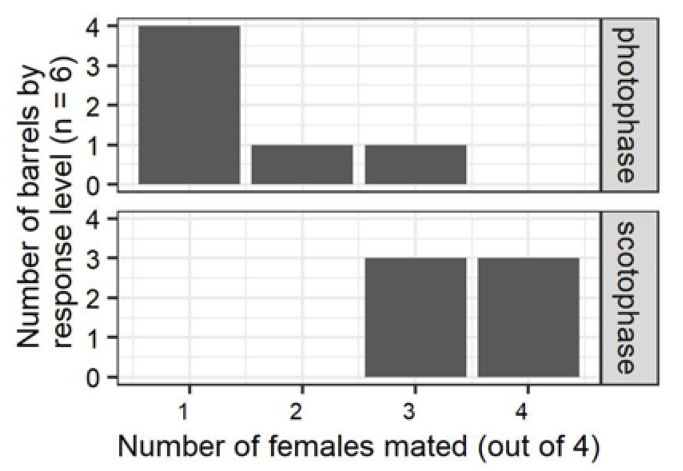
Proportion of females mated in 208 L chambers when treatment 5 (no mating disruption, see Figure 1) was performed during the photophase (above) and the scotophase (below). There was a significantly higher proportion of females mated when paired in scotophase than when paired in photophase (Fisher’s exact test, *n* = 12, *p* = 0.03).

**Figure 3 insects-11-00701-f003:**
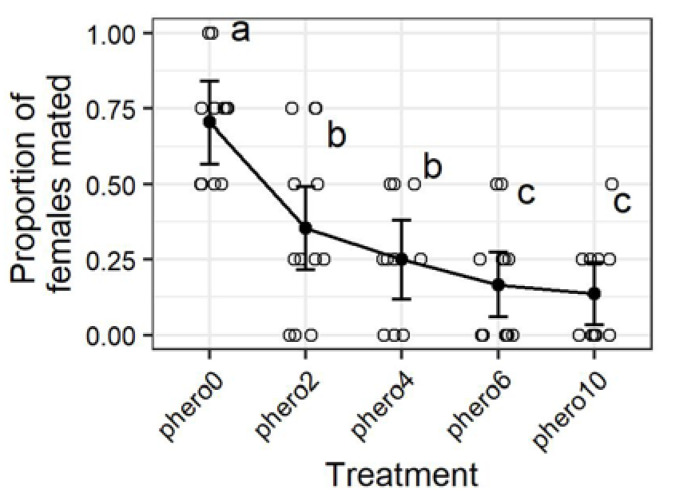
Females mated in 208 L chambers by mating disruption treatment (see Figure 1). Treatments phero0, phero2, phero4… etc. indicate the amount of time males were exposed to a pheromone mating disruption dispesnser; e.g., 2, 4, or 6 hours. Open circles represent individual observations, and the black points and whiskers respectively represent the mean and standard error. Means with different letters are significantly different (*p* < 0.05, GLM (generalized linear model) with binomial distribution).

**Figure 4 insects-11-00701-f004:**
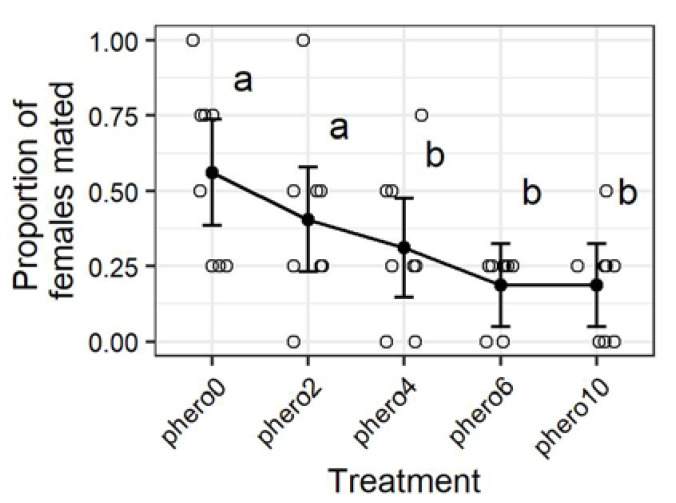
Females mated in 208 L chambers by mating disruption treatment (see Figure 1). Open circles represent individual observations, and the black points and whiskers respectively represent the mean and standard error. Means under different letters are significantly different (*p* < 0.05, GLM with binomial distribution).

**Figure 5 insects-11-00701-f005:**
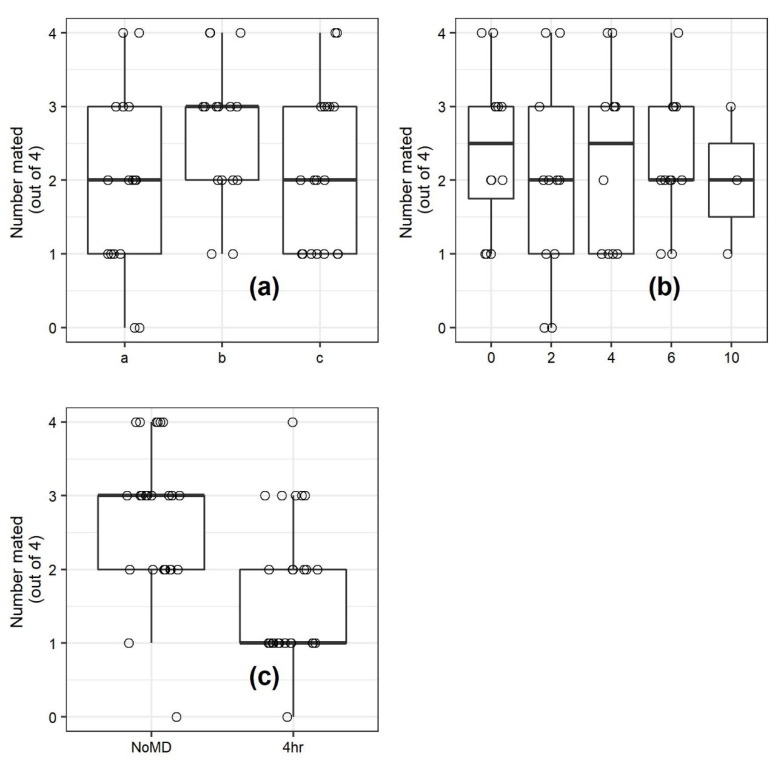
Effect of replicate in time (**a**), previous treatment (**b**), and prior exposure outside the chamber on mating in 208 L chambers (**c**). Males were held in a cage for the first 4 h of the scotoperiod and either exposed to a mating disruption dispenser (4 h) or not (NoMD), then transferred into the chambers. Females were introduced after six hours of scotoperiod. There was no significant effect of day (replicate in time) (χ^2^ = 3.37, df = 2, *p* = 0.18) or previous treatment (χ^2^ = 0.81, df = 4, *p* = 0.94. Effect of pre-exposure was, however, significant (generalized linear mixed model with binomial distribution, *F* = 12.07, df = 1, 43, *p* = 0.0012).

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
