# Peer review of "Persistence of Mating Suppression of the Indian Meal Moth Plodia Interpunctella in the Presence and Absence of Commercial Mating Disruption Dispensers"

_insects, 2020, doi:10.3390/insects11100701_

Round 1
Reviewer 1 Report
The knowledge about which are the mechanisms that govern mating disruption (MD) treatments is essential to optimize the implementation of these control tools. This manuscript gains insight on this issue by reporting a series of laboratory assays to evaluate the effect of commercial MD dispensers on the subsequent mating suppression of the stored-product pest Plodia interpunctella. The manuscript includes a centered introduction and the literature cited is appropriate.
In preliminary tests with containers of different sizes (28 mL, 470 mL and 208 L), authors evaluate if the arena is sufficiently large that mating occurs by response to the pheromone and not by chance. They conclude that the majority of the couples mated in the smaller containers (28 and 470 mL) and independently from the circadian rhythm. By contrast, couples inside the bigger containers (208 L) mate less frequently and, thus, they have to rely on the chemical signals (sex pheromone) to find mate. In this way, they validate the use of the barrels for the following assays: effect of the time and duration of the males’ exposure to MD dispensers inside the barrel, residual effect of the previous presence of dispensers and effect of males’ pre-exposure to MD dispensers outside the barrel. This is a very well planned series of experiments. The effects on mating that result from the different treatments are properly reported and support the conclusions of the work.
Manuscript is well written but there are many spelling mistakes throughout the text, please see the list below for some of them. Moreover, please clarify the volume of the second chamber: 470, 950, 946? These are mixed throughout the text. The same for the barrels, 208, 210 L?
I would recommend this work for publication after considering some minor aspects:
- Line 21: please replace ‘release’ with ‘dispensers’
- Line 22: delete ‘shared’ as you are not explaining here who is it shared with
- Lines 71-75: please rephrase, it is too long
- Line 77: replace ‘principle’ with ‘principal’
- Line 88: replace ‘exposes’ with ‘exposed’
- Line 101: part of the sentence is missing ‘Rolled bands of corrugated cardboard,’
- Line 104: delete parantheses after OAc
- Line 105: although it is unpublished I suggest briefly mentioning how the release rate was estimated, gravimetrically? gas chromatography?
- Line 116: delete ‘also’
- Line 133: insert ‘inside’ before ‘946 ml jars’
- Line 140: correct ‘cambers’
- Line 153: delete one of the ‘in’
- Line 157: correct ‘photoeriod’
- Line 164: correct ‘208 l’ not ‘ml’
- Line 213: Experiment 2 was conducted ‘within two to seven days of the previous experiment’, I guess you performed the experiments with all treatments (phero0, 2, 4…) on different days to eliminate the effect of the time passed (2-7 days). I mean, the residual pheromone emission from the barrels’ surface should be different whether 2 or 7 days have passed from the previous installation of the MD dispenser, did you check this effect? The variability observed in the data points of Figure 4 might be also due to this factor. I suggest adding a comment on this respect on the results, for example.
- Line 227: insert ‘.’ before ‘The 10 h treatment…’
- Line 231: replace ‘houses’ with ‘housed’
- Line 265: italize ‘P. interpunctella’
- Line 265: 950 ml arenas? or 470 ml?
- Line 305: delete ‘that that’ and insert ‘than’
- Line 334: 946 ml, please clarify volume
- Line 350: correct ‘compared’
- Line 355: delete one of the ‘in’
- Line 359: delete ‘1.8’
- Line 375: rephrase or replace ‘contact’ with ‘reaching the emission source’ or something similar
- Lines 369-393: please, revise this last paragraph. I cannot see a very clear sequence of examples or the order of the references cited is not adequate to properly connect the ideas with the conclusion of your experiments.
- Line 392: with ‘augmented’ you mean combining passive dispensers with microencapsulated formulations or aerosols? You say ‘in conjunction’ in the Abstract. Would this be economically affordable? Why not just shift to the other MD technologies? Have aerosols or microencapsulated sprays been tested and their efficacy reported for this pest? If not, you could say that based on the results of these experiments these other technologies are expected to work properly for mating disruption of P. interpunctella…
Author Response
See attached document

Reviewer 2 Report
The main question addressed by the research is rather interesting.
The work has some original aspect. It is comparable with other publications.
The paper well written and readable.
The conclusion of this paper matches the evidence presumable.
Instead of (Z9,E11)-tetradecadienyl acetate, please use
(9Z, 11E)-tetradecenyl acetate.
Author Response
The attachment duplicates the following response:
Thank you for getting us to take a closer look at the chemical nomenclature. The correct double bond placement is 9Z,12E. This is stated correctly on line 78 but incorrectly in the abstract, line 23. In both cases we have switched from Z9,E12 to 9Z,12E. This is a diene (two double bonds) and we believe that tetradecadienyl is more correct than tetradecynl. Our sources for verification are:
https://www.pherobase.com/database/compound/compounds-detail-Z9E12-14Ac.php
https://pubchem.ncbi.nlm.nih.gov/compound/5365642

Reviewer 3 Report
Comments from Reviewer
The authors tried to establish the new bioassay method for mating disruption, since the Indian meal moth is a sored product pest, which is different from other pest moth species in the field. Establishing the method is a good improvement for developing the pheromone dispensers for mating disruption. Other minor comments are as follows:
L.127: interal--->internal?
L.153: in in ---> in
L.227: time ---> time.
L.265: P. interpunctella ---> P. interpunctella (italics)
L.323: (A,B,C): uppercase are not found in the figure. (a)a,b,c are also confusing. Figure legend should be stand alone, so you should rewrite this for easy understanding.
L.332: use sex pheromone---> use of sex pheromone
L.344: However---> however?
L.355: in in ---> in
L.365: figs. ---> Figs.
L.448: Cydia pomonella ---> Cydia pomonella (italics)
Author Response
Suggestions incorporated. See attached response

Reviewer 4 Report
The paper is written well. There are only minor editorial corrections (see below). The experiments were designed well and data analyzed correctly. The authors examined competitive and non-competitive mating suppression via 4 experiments. I would have liked to see their statistics especially ANOVA results where they show block and treatment effect. These data need to be presented in Results. Also I could not access Tables 1 and 2, so I cannot comment on them. The link to tables says "File not found".
Since the authors had so many replications, instead of just examining bursa copulatrix for spermatophore bundle, they could have counted the number of spermatophores in bursa. Indian meal moth males are capable of mating with 6 females, and a female can receive 10 males. However, to lay full complement of eggs only one mating is necessary. Counting the spermatophore numbers would have indicated if the treatments also prevented more than one mating. In addition, they could have allowed moths in half od the replicates lay eggs and the authors could have counted number of eggs laid and number of eggs that hatched. This could have have shown benefits of mating disruption besides just whether the female has mated or not. To me these aspects would have added more value to the paper.
Here are minor editorial changes:
Line 14. Throughout the paper either use 2 hr or two hours or 4 hr or four hours and not both to be consistent.
Line 15. Delete comma after that.
Line 28. Change dose to concentration as authors do not know exactly how much pheromone each male was exposed to.
Line 32. Delete likely an.
Line 43. Add s after semiochemical, and add period after [3].
Line 44. Start with Other strategies....
Line 45. Change are to is.
Line 53. Delete an; add s after factor.
Line 72. Add period after [29].
Line 73. Start with Response....
Line 77. Change principle to principal.
Line 84. Change l after 208 to L throughout the paper.
Line 87......add with males exposed and unexposed all night. Delete other words.
Line 92. add on after information.
Line 97. Change founded to found.
Line 101. change photobase to photophase, same line starting Rolled bands of corrugated cardboard of xx cm long and xx cm wide were placed above the diet to serve as a pupation site. The sentence is abruptly cut off.
Line 122. UJse abbreviations for states, like ID for Idaho, USA. Elsewhere CA for California. Make use USA is mentioned as well.
Lines 134-135. Adults 2-4 days after eclosion were used in experiments. Delete other words.
Line 140. Add h to cambers and delete prior.
Line 144 add comma after CA.
Line 157. Add p to photoperiod.
Line 221. Change 210 to 208 L.
Line 227. Give space betwen The and 10 h. Here you said h but other places you used hr. Be consistent.
Line 228. Change was to were after There...
Line 231. Change houses to housed.
Line 233. Change California to CA add comma after CA.
Line 235. Give units for dimensions.
Line 237. Give units for dimensions.
Line 265. Italicize P. interpunctella. Did not see S1 and S2 Tables.
Line 285. N should be italicized.
Line 289. Z and P should be italicized.
Figs 3 and 4 should be presented as bar graphs and not as line graphs. Just give mean and SE.
The a, b, c, are actually are for means and not reps. Need F, df, and P values in text for blocks and treatments.
Line 299. Change under to with.
Line 304. Italicize Z and P.
Line 326. Change were to was.
Line 329. Give space between 1 and 43.
Line 351. How does re-emission work in real world settings?
Line 359. 1.451.8 meter square? Correct this.
Line 371. Delete the oriental fruit moth and (Busck) as these were mentioned before.
